# Precision or Personalized Nutrition: A Bibliometric Analysis

**DOI:** 10.3390/nu16172922

**Published:** 2024-09-01

**Authors:** Daniel Hinojosa-Nogueira, Alba Subiri-Verdugo, Cristina Mª Díaz-Perdigones, Alba Rodríguez-Muñoz, Alberto Vilches-Pérez, Virginia Mela, Francisco J. Tinahones, Isabel Moreno-Indias

**Affiliations:** 1Instituto de Investigación Biomédica de Málaga y Plataforma en Nanomedicina—IBIMA Plataforma BIONAND, 29590 Malaga, Spain; alb.sub@gmail.com (A.S.-V.); cmaria.diaz.sspa@juntadeandalucia.es (C.M.D.-P.); albarodriguezmunoz9@gmail.com (A.R.-M.); alberto_v4@hotmail.com (A.V.-P.); virginia.mela@ibima.eu (V.M.); isabel.moreno@ibima.eu (I.M.-I.); 2Unidad de Gestión Clínica de Endocrinología y Nutrición, Hospital Universitario Virgen de la Victoria, 29010 Malaga, Spain; 3Department of Medicine and Dermatology, Faculty of Medicine, University of Málaga, 29010 Malaga, Spain; 4CIBER Fisiopatología de la Obesidad y Nutrición (CIBERObn), Instituto Salud Carlos III, 28029 Madrid, Spain

**Keywords:** personalized nutrition, precision nutrition, bibliometric analysis

## Abstract

Food systems face the challenge of maintaining adequate nutrition for all populations. Inter-individual responses to the same diet have made precision or personalized nutrition (PN) an emerging and relevant topic. The aim of this study is to analyze the evolution of the PN field, identifying the principal actors and topics, and providing a comprehensive overview. Therefore, a bibliometric analysis of the scientific research available through the Web of Science (WOS) database was performed, revealing 2148 relevant papers up to June 2024. VOSviewer and the WOS platform were employed for the processing and analysis, and included an evaluation of diverse data such as country, author or most frequent keywords, among others. The analysis revealed a period of exponential growth from 2015 to 2023, with the USA, Spain, and England as the top contributors. The field of “Nutrition and Dietetics” is particularly significant, comprising nearly 33% of the total publications. The most highly cited institutions are the universities of Tufts, College Dublin, and Navarra. The relationship between nutrition, genetics, and omics sciences, along with dietary intervention studies, has been a defining factor in the evolution of PN. In conclusion, PN represents a promising field of research with significant potential for further advancement and growth.

## 1. Introduction

Currently, one of the most significant challenges facing the global food system is food and nutrition insecurity [1]. The mission is to ensure that individuals maintain an adequate nutritional status, in addition to the sustainability of the food system [2]. In this regard, organizations such as the European Commission and the United Nations support strategies promoting a healthier and more sustainable food system [3,4].

Dietary recommendations are formulated on a population level, taking into account average requirements according to model groups classified by age, gender, region or lifestyle [5,6,7]. However, within each demographic group, there is a considerable range of responses to different dietary patterns [5,7]. The heterogeneous dietary responses of individuals are the result of a complex interplay of factors, including genetic, physiological, and phenotypic determinants; background medical history; lifestyle practices, such as physical activity, sleep routines, or dietary habits; and socio-economic and cultural factors, such as the food environment, educational level, ethical beliefs, consumer behavior, and culinary practices [8,9,10]. These factors highlight the need to not only develop generalized recommendations, but also to advance the development of more individualized nutrition to enhance individual responses, which is referred to as precision or personalized nutrition (PN).

The objective of PN is to improve health and prevent, manage, and treat diseases through the development of nutrition approaches, by using multidimensional data for each individual [9,11,12,13,14]. In recent years, food science and technological advances have greatly improved, allowing for the accurate quantification of a greater number of variables at the individual level. This has enabled a more nuanced understanding of the specific needs of each individual [6,11,15]. In PN, there can be different levels of personalization, ranging from studies that examine only dietary and lifestyle implications, as is typical of general recommendations, to more complex studies that analyze genetic or gut microbiome factors [16,17]. It is crucial to recognize that the more factors are analyzed, the greater the understanding of the possible interactions that occur [18], and the more effectively PN can be created [15]. For instance, the field of molecular biology has facilitated a deeper comprehension of the manner in which nutrients influence metabolic response [5,7,9]. In this sense, the potential of omics sciences, such as nutrigenomics or proteomics, has become evident [6,19]. Furthermore, the reduction in costs and the increase in the affordability of certain techniques, such as gut microbiota analysis, have facilitated their more frequent application in nutrition studies [19,20,21]. 

Another crucial aspect of PN is the evolution of technologies such as big data analysis, artificial intelligence (AI), the use of wearable devices, and the increase in mobile applications (APP) that have facilitated the evaluation and monitoring of diets, enabling the analysis of a greater amount of data, even in real time [22,23,24]. This has enabled PN to integrate traditional analysis with a large volume of data, thereby facilitating the more precise characterization of the individual response to diet and the formulation of more effective personal nutrition advice and treatment [24].

Consequently, the convergence of these multiple factors is enabling the advancement of PN, thereby facilitating the availability of more data to individuals, industry, and relevant organizations to improve diet quality and health status from a more personalized point of view.

In spite of the great potential of PN, this phenomenon is still relatively young, albeit with great interest on the part of the different actors involved. Awareness of the principal countries, institutions, authors, and trends associated with PN can facilitate the incorporation of other researchers and organizations. The aim of this study is to examine all the existing scientific research on PN through a bibliometric analysis. This will permit the identification of publication trends, the principal actors, and the levels of collaboration, thereby affording a comprehensive overview of this field, which will facilitate a more profound understanding of its current state and contribute to its future advancement.

## 2. Materials and Methods

### 2.1. Scientific Publications Search Strategy

The present study employs a bibliometric methodology to examine existing scientific research about PN [25]. A comprehensive search and compilation of all relevant research was conducted using the Web of Science (WOS) database, which is considered to be one of the most influential and up-to-date sources of bibliographic data on scientific research [26]. In order to identify all existing documents, a literature search was conducted from its first occurrence (1997) up to the end of June 2024. The following keywords were employed in the document search: “personalised nutrition” OR “precision nutrition” OR “personalized nutrition”. The “All Fields” filter was applied to identify the largest number of documents containing these keywords. The search yielded a total of 2148 potentially relevant documents. All available information was collated, including the title, year of publication, abstract, keywords, authors, journals, and other relevant elements. All data were extracted in the. txt format, because of its versatility, for further analysis.

### 2.2. Data Analysis

The number of publications per year was determined by collecting all information within the analyzed period (1997–2024). In order to delimit the most relevant information, several selections were considered. Regarding scientific field categories, the 15 most pertinent categories from the WOS database were selected. Of the initial 99 countries considered, only the 15 most relevant were selected, based on the number of publications and citations. The same methodology was employed for authors (*n* = 8730) and journals (*n* = 722), with the exception that for authors, a minimum of 5 publications was required. For the institutions (*n* = 2828), the 20 most cited with at least one article were selected. Furthermore, from a total of 8476 keywords, 50 keywords with the highest number of occurrences and a minimum of 10 occurrences were selected. Finally, from a total of 40,342 words in the title, source, or abstract, the 50 words with the highest frequency of occurrence were subjected to a similar analysis.

The years of publication, languages, publishers, WOS categories, and open access status were analyzed using the online analysis platform provided by WOS. The bibliometric analyses for the rest of the data set were performed with VOSviewer 1.6.20 software, a software tool for constructing and visualizing bibliometric networks [27]. Data were processed and presented as percentages, unless otherwise stated. 

## 3. Results

The bibliometric analysis revealed that 98% of the 2148 publications were in English, with the remaining 2% in other languages. Furthermore, only 67.6% of the publications were open-access. Figure 1 presents the distribution of the publications by year and country, as well as the highest numbers of publications and citations, according to the publications identified up to June 2024.

The distribution by year revealed that the first appearance of the term PN was in 1997, followed by an upturn in the period between 2007 and 2013. Thereafter, an exponential growth occurred from 2015, peaking near 350 manuscripts per year. When focusing on the countries, the USA, Spain, and England were identified as the main contributors in terms of both the number of articles and the number of citations. The majority of the countries exhibited a comparable distribution in terms of both the number of publications and the number of citations, although the rankings differed due to the varying contributions of each country. Countries such as Greece, France, India, and Mexico were represented by high numbers of documents, while countries such as Sweden, Israel, Denmark, and Belgium were represented by high numbers of citations.

A total of 131 WOS scientific categories were identified. The 15 most important categories, as determined by their relative importance, are presented in Table 1. Nutrition Dietetics stands out from the rest, with almost a third of the published manuscripts. This is followed by Food Science Technology (9.1%), which, together with Endocrinology Metabolism, Genetics Heredity, and Biochemistry Molecular Biology, is the most representative, although it is far from the Nutrition and Dietetics rate. The remaining categories exhibit a lower level of representation. 

In terms of the distribution of publications among the different publishers, Elsevier leads with 19% of the total, followed by MDPI, with 14.4%, and Springer Nature, with 10.9%. Frontiers Media SA, Wiley, Karger, Oxford University Press, Cambridge University Press, and Taylor & Francis have approximately 7% to 4% of the total number of publications. The remaining publishers collectively account for 24% of the total. A more detailed examination of the data reveals that there are differences in the distribution of the publications by journal, as measured by both the number of citations and the number of documents (Table 2). Furthermore, it can be observed that the mean number of citations per article is 19.3.

The most frequently cited universities and research institutions in the field of PN studies were identified through a collaboration network analysis. Collaborations between institutions and their temporal context were identified (Figure 2).

Some institutions, such as the University of Navarra, the University of Newcastle, University College Dublin, the University of Reading, and the Carlos III Health Institute, are highly relevant to the field of PN. In addition, other institutions, such as Tufts University, the University of Copenhagen, Deakin University, and the University of Laval, are emerging and gaining in relevance. Taking this into account, an analysis of the most relevant authors was carried out (Table 3).

Of the 50 most cited authors, 56% are male and 44% are female. If we examine the top 15 authors in terms of number of articles and number of citations, there are authors who appear in both categories, such as Brennan, Lorraine, Alfredo Martinez, J., Ordovas, Jose M., and Mathers, John C.

Finally, a co-occurrence and cluster analysis of the keywords (Figure 3) and of the words that appear in the title, source, abstract of the publications (Figure 4) was carried out.

The co-occurrence analysis of the keywords was classified into three large clusters, as shown in Figure 3, which illustrates the trends in the PN. The five most relevant words in the green cluster are “metabolic syndrome”, “body mass index”, “Mediterranean diet”, “cardiovascular disease”, and “polymorphisms”. The five most relevant words in the orange cluster are “personalized nutrition”, “precision nutrition”, “diet”, “nutrition”, and “health”. The five most relevant blue cluster words are “obesity”, “prevention”, “weight loss”, “intervention”, and “adults”.

The other co-occurrence analysis of the most cited words yielded two main clusters, as illustrated in Figure 4. It is evident that one group can be identified as comprising the dietary interventions and characteristics of the subjects, while another group can be identified as comprising technology and analysis. The most relevant words in the orange cluster are “microbiota”, “research”, “development”, “application”, and “review”; the most relevant words in the green cluster are “group”, “participant”, “change”, “association”, and “risk”.

## 4. Discussion

The practice of personalized or precision nutrition is currently growing in popularity as one of the best ways to improve the health of individuals. After the development of the population-based nutrition approach, with its huge advancements for the welfare state, it has become evident that not one size fits all. In this regard, nutrition needs to focus on the individual.

According to the scientific research found in WOS, the first mention of PN was in 1997 and referred to animal nutrition [28]. Two years later, in 1999, PN was first applied to humans, in terms of education [29]. In fact, PN began to emerge between 2004 and 2006, when the first links between nutrition and diet, genetics, and other omics sciences were discussed [19,30,31,32,33,34]. Subsequently, PN became increasingly well known, and in the period between 2007 and 2013, the field experienced a major resurgence, coinciding with significant milestones in genetics and the omics sciences [35,36], with 2015 as the date when PN began to grow exponentially (Figure 1). The large variability in the dietary responses among the population was the event that triggered this scientific interest [37], while the development of the field was made possible because all omics scientific methodologies became more affordable [36,38]. In 2016, the International Society of Nutrigenetics/Nutrigenomics (ISNN) provided one of the most widely used definitions of PN [16]. In essence, PN can be defined as the process of identifying the optimal response to nutrient intake for an individual, taking into account the interactions between metabolic, environmental, social, and genetic factors. In contrast to the conventional notion of PN, which entailed adapting dietary regimens to suit individual needs and preferences, the advent of omics sciences has introduced a new dimension to this field. The integration of omics technologies, including foodomics, nutrigenomics, epigenomics, transcriptomics, metabolomics, and metagenomics, has led to a paradigm shift in the understanding and application of PN. Furthermore, three levels of PN action were distinguished: (1) nutrition based on general guidelines for population groups, (2) individualized nutrition that aggregates phenotypic information, and (3) genotype-driven nutrition. The final objective was to integrate all the available information in order to ensure the most appropriate individual response, given the circumstances.

Furthermore, the first population-based personalized nutrition projects, such as the Food4Me study, started during this period [12,37]. The Food4me project was a landmark project led by Professor Mike Gibney, University College Dublin, and marked the beginning of the first PN studies. It was a clinical trial involving participants from several European countries and marked an important milestone in the practical application of PN [12,39]. The study’s importance is reflected in the great relevance of the authors participating in it, such as Mathers, John C., who is one of the authors with the most publications and citations on PN (Table 3). It is also reflected in Figure 2, which shows the importance of University College Dublin and many of the links with the institutions involved in the project consortium. The significance and potential of the field are evident since establishment and throughout its history, organizations such as the European Commission have prioritized the development of personalized nutrition, providing financial support for research projects like the Preventionomics study [40], the Protein Project [41], and the Stance4Health Project [42].

Up to 2022, the growth of PN research progressed exponentially, but as can be seen in Figure 1, the number of publications decreased in 2023. We will have to wait a few years to know whether this year represents an inflection point in the number of PN papers published, or whether it was only a slight decrease in the number of manuscripts, something that has been observed in many other fields after the impact of the COVID-19 pandemic, when scientific research suffered from a general increase in [43]. 

In terms of countries, the USA, Spain, and England lead PN research in terms of both the number of publications and the number of citations. This may be related to the progress made by the United States and Spain in the fields of omics and dietetics [18,24,44,45,46,47,48,49,50]. In the case of England, this could be related to the presence of important research projects on PN, such as the ZOE PREDICT 1 study [11,51]. As can be observed in Figure 1**,** China, Canada, and other European countries such as the Netherlands, Germany, and Italy, also have a presence in the field of PN. In terms of the number of publications, it is possible to identify countries such as India and Mexico, which demonstrate a commitment to omics sciences in PN and an increase in collaborative networks [45,52,53]. The analysis of the number of citations indicates that countries such as Israel and Sweden exert a considerable influence on PN, through either relevant publications or collaborations within the field [54,55].

On the other hand, the bibliometric analysis reveals that Nutrition & Dietetics is the most representative scientific category in PN, followed by Food Science Technology, which is understandable given the nature of the field (Table 1). In addition, the fact that Endocrinology Metabolism, Genetics Heredity, and Microbiology have a higher percentage than other disciplines is fully justified, given that these areas of research are key factors in PN. As shown in Figure 3, several of these factors can be identified, including nutrigenetics, genomics, epigenetics, the microbiome, and the metabolome. This highlights the crucial need to comprehend the individual at all levels for personalized nutrition [19,21,34,44,45,46], including factors that are increasingly important, such as the gut microbiota and its interactions with diet composition and host metabolism [56]. Indeed, the fact that researchers in the Microbiology field have paid so much attention to PN is indicative of the importance of the microbiota for the development of PN.

Concerning to the role of publishers, it is noteworthy that three publishers (Elsevier, MDPI, and Springer Nature) have the greatest weight (44.3%) in terms of PN publishing, which is also related to the volume of manuscripts published by these publishers yearly. Regarding journals, it is noteworthy that a number of them appear among the 15 most relevant based on both the number of publications and the number of citations. Examples include *Nutrients*, as well as *Frontiers in Nutrition* and *Genes and Nutrition* (Table 2). Additionally, a few journals stand out for having a high number of citations, despite having a relatively low number of publications. These include *Cell* (*n* = 2), *Nature* (*n* = 1), and *Circulation* (*n* = 2). This phenomenon is observed across a wide range of scientific fields [57,58,59,60].

In the network analysis presented in Figure 2, certain institutions stand out, including Newcastle University, University College Dublin, and the University of Navarra, which are highly interconnected, suggesting a high frequency of collaboration between them, and they are presented as the main nodes in PN research. Other examples are Tufts University and the University of Copenhagen, which are connected to several institutions but seem to have more specific collaborations. The timeline reveals a notable increase in the intensity of collaborations during the years of 2017 and 2018. Concurrently, institutions such as Deakin University and Laval University have emerged as significant contributors to the field of PN. This phenomenon coincides with an increase in the visibility of Australia and Canada as relevant countries, both in terms of the number of citations and the number of publications (Figure 1).

Table 3 presents a classification of the most relevant authors, some of whom have both a high number of publications and a high number of citations received. These authors have been and continue to be pioneers in the field of PN, either through their participation in projects or through their contributions to high-impact publications. It is important to highlight the contributions of authors such as Brennan, Lorraine, Alfredo Martinez, J., Mathers, John C., and Ordovas, Jose M., who have participated and continue to participate in PN projects such as Food4Me, The Food Biomarker Alliance (FOODBALL Project), and other dietary interventions focused on PN [37,61,62]. In addition, some authors should be highlighted who, even with fewer publications, have a significant impact on the field of PN. These include Elinav, Eran, Segal, Eran, and Zmora, Niv [55,63,64].

The co-occurrence network analysis of the keywords (Figure 3) was employed to visualize and explore the relationships between different concepts. Three clusters were distinguished, with the terms “personalized nutrition” and “precision nutrition” situated in the middle of the graph, acting as connecting nodes that link these groups. The green group is concerned with the molecular and genetic aspects of PN, including the various genetic and epigenetic factors that influence metabolic diseases and nutrient response. The blue cluster illustrates the interactions between genetics and polymorphisms with obesity, as well as the relationship with physical activity and disease prevention. This cluster places considerable emphasis on PN-based dietary interventions as tools in the fight against obesity and in support of weight loss. The orange cluster examines the relationship between precision nutrition and its impact on health through diet and the gut microbiota. It focuses on how dietary components and supplements can affect health by modulating the microbiota and metabolism.

The analysis presented in Figure 4 reveals the existence of two major clusters. The green cluster is primarily concerned with studies and clinical trials in the field of nutrition. The cluster includes terms describing the characteristics of study participants, the composition of trial groups, changes in parameters such as weight and body mass index, as well as dietary intake and participant genetics. In contrast, the orange cluster is primarily concerned with research and development aspects, as well as the microbiota. In addition, the cluster includes terms describing scientific research, such as literature reviews, product development, and applied technologies. Furthermore, the category includes concepts such as nutrigenomics and metabolomics. It is notable that the relationship between these two clusters is principally defined by the concepts of “microbiota”, “trial”, and “change”. This suggests that research into the microbiota is closely related to PN studies and the changes observed in participants. This relationship also emphasizes the role of genotyping in the practical implementation of nutritional interventions and their impact on human health, gut microbiome modulation, and metabolism.

Among the findings, it is notable that dietary interventions represent significant examples of the potential applicability of PN. The implementation of PN in dietary interventions has been demonstrated to have a considerable impact on different population groups. In fact, one of the approaches proposed within the framework of PN is the implementation of subject-to-subject nutritional interventions. This method has been proposed as one of the most effective for promoting change and yielding outcomes in specific population groups. Indeed, once the methodology has been developed, it can be applied to a larger population [65].

For example, the use of PN has increased in the general population, notably due to the implementation of diverse research projects and the integration of innovative technological tools, such as mobile applications that provide users with real-time and continuous monitoring [66]. The use of PN is also being extended to other populations, such as children, for whom adequate nutrition is essential for growth and development. The application of PN facilitates the adaptation of recommendations to provide optimal nutritional support for this population [67]. Another example to be considered is that of pregnant women, for whom the presence of PN could be considered essential to ensure the health of both the mother and the fetus [68]. PN can also assist athletes in optimizing their performance and recovery through the provision of diets that are tailored to their specific requirements, thus facilitating the achievement of optimal physical condition [69].

In the context of pathology, PN has demonstrated considerable potential in the management of chronic conditions such as obesity and diabetes, where it has been most extensively deployed. In the context of obesity, the primary objective is the reduction in and control of body weight [70,71]. In the case of diabetes, the principal focus is the regulation of glucose levels through the incorporation of omics technologies [71]. Furthermore, the implementation of PN has been demonstrated to confer benefits with respect to cardiometabolic health. By considering factors such as genetics, the microbiome, and lifestyle, it provides dietary strategies that reduce the risk or improve the management of these diseases [72]. Similarly, in the context of cancer treatment, the role of PN in disease management is also of significant importance. The objective is to assist patients in the maintenance of an adequate nutritional status throughout the course of their oncology treatments. This is a crucial factor in improving both the tolerance of the treatment and the quality of life of the patient [73]. The efficacy of PN has also been demonstrated in the treatment of other pathologies, particularly through the consideration of factors such as the gut microbiota when modifying diets to align with individual requirements, thereby enabling more appropriate management of these conditions [74,75].

In order for these interventions to be effective, it is essential that users are provided with the appropriate nutritional information, thus enabling them to make well-informed decisions regarding their diet [22]. Consequently, PN prioritizes not only the adequate intake of nutrients, but also the implementation of strategies to facilitate dietary behavioral change [76], including personalized meal planning, continuous monitoring of dietary habits, and education on making more informed food choices. 

One of the principal advantages of PN is the level of customization that it offers. The setting of different levels is possible even within the same study, although this is contingent on the availability of resources and the desired outcomes. In this context, research in which different levels of PN are conducted can assist researchers in the identification of the most significant methodologies in order to take advantage of their resources [77].

Overall, the results of the studies discussed above demonstrate the potential efficacy of PN dietary interventions, with no significant safety concerns. Nevertheless, the current evidence base is limited by small sample sizes and a lack of comparability between studies. To obtain robust evidence and ensure comparable results, larger and longer trials with standardized protocols are required [78]. Therefore, PN in dietary interventions represents an effective solution for the promotion of health and the prevention of disease, offering tangible benefits at both individual and population levels. For example, the article by San-Cristobal et al. emphasizes the importance of precision nutrition in the development of individualized and effective treatment approaches, considering the impact of macronutrient intake on genotype and microbiome [79]. Another example can be found in the article by Picó et al., which emphasizes the necessity of identifying biomarkers that can be employed to analyze physiological or pathological responses to specific food components or diets, discern inter-individual variations to facilitate the formulation of personalized dietary recommendations, and, thus, contribute to the advancement of PN [80]. According to our results, advances in these fields are expected in the near future.

In this study, it is also important to highlight the role that the development of technology and omics sciences has played in the context of PN, as this has been identified as a focal point. Currently, there is a notable advancement in omics techniques [81,82,83], including their analysis and interpretation [84]. Additionally, the integration of artificial intelligence (AI) is becoming increasingly prevalent across a range of application areas, such as food omics [85]. In addition, there is a growing interest in the use of biomarkers of food intake as a means of improving the accuracy of dietary surveys, and in the use of sensors to capture more data and contribute to the development of more accurate and effective dietary interventions [86,87]. Most recently, the increasing prevalence of the use of apps and smartphones as tools has transformed the methodology and conceptualization of PN [24,64,88]. Although there are still some gaps to be addressed, such as the need for more evidence and clinical scalability and the effective integration of omics data, ethical and legal challenges, high costs, and the limited accessibility to omics technologies, PN offers a wide range of opportunities for disease prevention and health improvement [89]. These include personalized diets, the development of new functional foods, the use of advanced technologies, and specific nutritional interventions to improve public health [40,42,88,89].

There were certain limiting factors inherent in this study. Although WOS is one of the largest databases of scientific research [26], other databases, such as Scopus, PubMed, and Embase, were not utilized in this study and could have increased the amount of relevant information. However, WOS was selected for its versatility in information extraction and examination, in addition to being one of the most complete databases for the development of this kind of analysis. Moreover, although special care has been taken, it is possible that certain extracted metadata may have errors or may not be well standardized, which is common in bibliometric studies [90]; nevertheless, complex data were not used and checks were made to avoid potential biases.

The future of PN appears to be promising in the short, medium, and long term, driven by technological advances. The precision and accessibility of omics sciences will continue to improve, thereby facilitating their implementation on a more frequent basis [91]. Moreover, the data will be incorporated into mobile applications and online platforms, which will facilitate the generation of more reliable and personalized nutritional recommendations [92]. On the other hand, foods may be developed in order to provide “cellular nutrition” through a diet that includes bioactive components that regulate gene expression in response to the individual’s nutritional requirements [93]. Additionally, at the level of food production, the use of 3D printing and cell-cultured foods could facilitate the creation of highly customized foods, which are specifically designed with consideration of the nutritional requirements and taste preferences [94,95,96].

The continuous advancement of artificial intelligence and machine learning will permit the development of more sophisticated algorithms with the capacity to analyze ex-tensive data sets and correlate them with other variables, providing optimal dietary recommendations in a continuous and dynamic manner [97]. Furthermore, the use of advanced AI assistants and an increased number of sensors or augmented reality devices may facilitate continuous guidance on dietary choices [98,99].

One of the most significant challenges will be to disseminate these developments and integrate PN into the standard treatment of chronic diseases and health promotion in hospitals and other healthcare centers [47,100]. However, PN will also present certain risks, including novel ethical challenges, particularly those pertaining to data protection [101]. Furthermore, disparities in access to PN may emerge between populations, since individuals may have varying levels of access to these technological advancements [102]. Additionally, there is a potential risk of dietary uniformity on a global scale, which could result in a reduction in the diversity of traditional foods and their associated cultural impact [103,104].

## 5. Conclusions

Personalized nutrition represents an innovative approach that is emerging as a significant area of interest, with notable growth in research activity, especially in recent years. The role of different organizations in supporting significant research projects at the vanguard of this field and their impact on its development have been focal points within this field. Furthermore, evidence indicates that particular countries, along with their associated institutions, exert a significant influence within this field, with a notable emphasis on specific forms of PN, such as the future of dietary therapy. A number of scientists and publications have made a substantial contribution to the field of PN, promoting its dissemination and expansion. Another noteworthy strength is the significant collaborative effort within the scientific community to advance the field. It is also important to highlight how, through the application of omics sciences and technological advances, PN has advanced and continues to contribute to what could be considered a nutritional revolution in terms of developing and implementing individual recommendations. The future of PN is expected to be characterized by its integration into the daily lives of individuals, with the potential to personalize dietary and health-related choices at the individual level. However, these developments will also give rise to a number of ethical, social, and environmental challenges that will require careful consideration.

## Figures and Tables

**Figure 1 nutrients-16-02922-f001:**
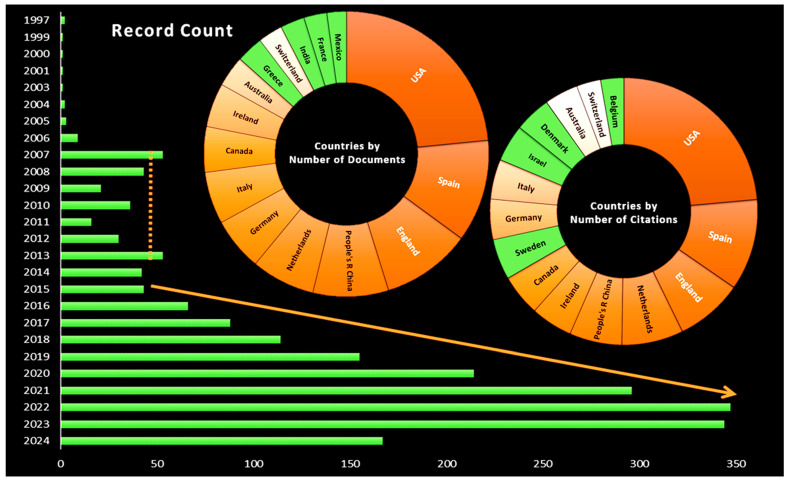
Number of articles published per year, between 1997 and 2024, and the distribution of countries with the highest number of publications and the highest number of citations. The countries indicated in green are those that have been identified to be different between the two categories.

**Figure 2 nutrients-16-02922-f002:**
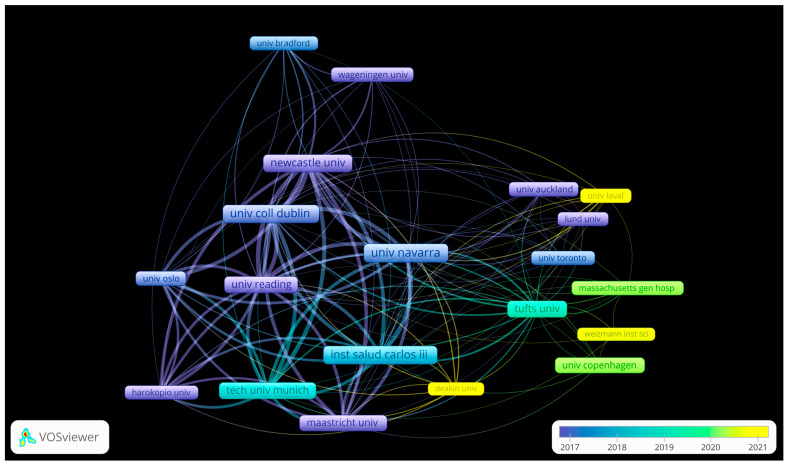
Exploration of the most relevant institutions according to the number of citations and their collaboration network in a temporal context.

**Figure 3 nutrients-16-02922-f003:**
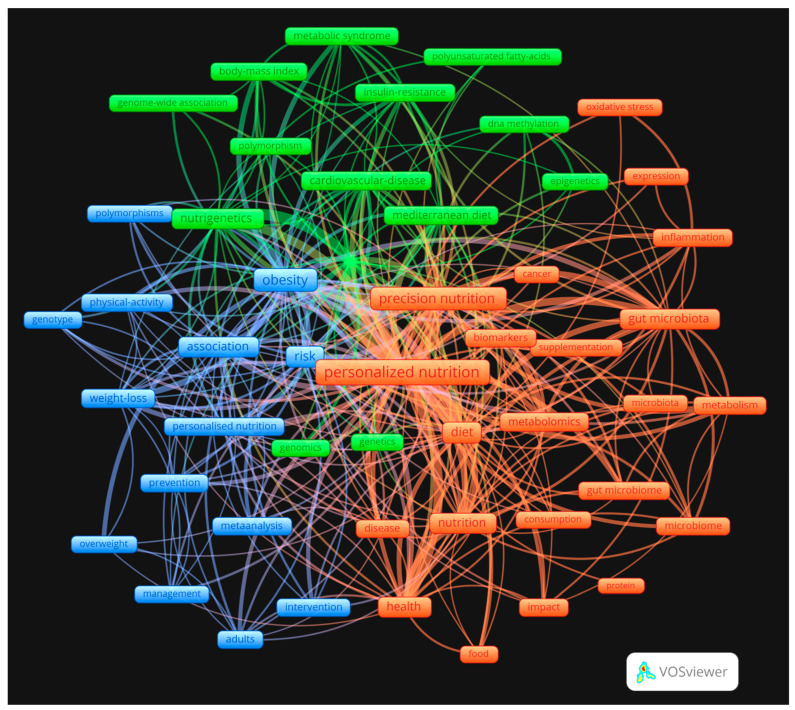
Co-occurrence analysis of the total keywords.

**Figure 4 nutrients-16-02922-f004:**
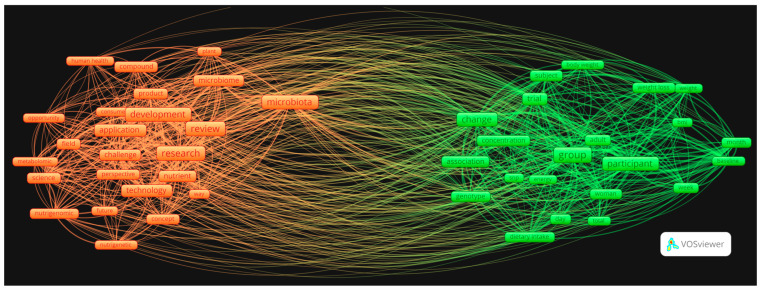
Co-occurrence analysis of the words that appear in the titles, sources, and abstracts of the publications.

**Table 1 nutrients-16-02922-t001:** Percentage distribution of the 15 most relevant WOS categories of PN manuscripts published between 1997 and June 2024.

Web of Science Categories	% Count	Web of Science Categories	% Count
Nutrition Dietetics	32.8	Health Care Sciences Services	1.5
Food Science Technology	9.1	Public Environmental Occupational Health	1.4
Endocrinology Metabolism	5.5	Agriculture Dairy Animal Science	1.4
Genetics Heredity	4.9	Biotechnology Applied Microbiology	1.4
Biochemistry Molecular Biology	4.3	Pharmacology Pharmacy	1.4
Medicine General Internal	1.7	Gastroenterology Hepatology	1.3
Microbiology	1.6	Cell Biology	1.3
Multidisciplinary Sciences	1.6	Others Categories	29.1

**Table 2 nutrients-16-02922-t002:** Distribution of journals according to the total number of publications and total number of citations.

By Number of Documents	By Number of Citations
*Nutrients* (*n* = 171)	*Nutrients* (*n* = 3131)
*Frontiers in Nutrition* (*n* = 96)	*Cell* (*n* = 1538)
*Genes and Nutrition* (*n* = 67)	*Nature* (*n* = 1393)
*Proceedings of the Nutrition Society* (*n* = 60)	*Circulation* (*n* = 1155)
*Annals of Nutrition and Metabolism* (*n* = 58)	*Frontiers in Nutrition* (*n* = 1126)
*American Journal of Clinical Nutrition* (*n* = 38)	*Gut* (*n* = 1030)
*Advances in Nutrition* (*n* = 36)	*American Journal of Clinical Nutrition* (*n* = 1012)
*Critical Reviews in Food Science and Nutrition* (*n* = 26)	*Genes and Nutrition* (*n* = 981)
*Journal of Nutrition* (*n* = 24)	*Trends in Food Science and Technology* (*n* = 980)
*Molecular Nutrition and Food Research* (*n* = 22)	*Journal of the Academy of Nutrition and Diet* (*n* = 873)
*Nutrition Reviews* (*n* = 21)	*Advances in Nutrition* (*n* = 866)
*International Journal of Molecular Sciences* (*n* = 20)	*Proceedings of the Nutrition Society* (*n* = 769)
*Foods* (*n* = 20)	*Molecular Nutrition and Food Research* (*n* = 689)
*BMJ Nutrition, Prevention and Health* (*n* = 19)	*Journal of Nutrigenetics and Nutrigenomics* (*n* = 650)
*Trends in Food Science and Technology* (*n* = 18)	*PLoS ONE* (*n* = 557)

**Table 3 nutrients-16-02922-t003:** Distribution of authors according to the total number of publications and total number of citations.

By Number of Documents	By Number of Citations
Brennan, Lorraine (*n* = 50)	Elinav, Eran (*n* = 2198)
Alfredo Martinez, J. (*n* = 43)	Segal, Eran (*n* = 1769)
Mathers, John C. (*n* = 42)	Mathers, John C. (*n* = 1687)
Gibney, Eileen R. (*n* = 39)	Ordovas, Jose M. (*n* = 1670)
Navas-Carretero, Santiago (*n* = 38)	Zmora, Niv (*n* = 1644)
San-Cristobal, Rodrigo (*n* = 38)	Ben-Yacov, Orly (*n* = 1589)
Ordovas, Jose M. (*n* = 37)	Lotan-Pompan, Maya (*n* = 1588)
Daniel, Hannelore (*n* = 35)	Weinberger, Adina (*n* = 1588)
Fallaize, Rosalind (*n* = 35)	Rein, Michal (*n* = 1587)
Lovegrove, Julie A. (*n* = 35)	Brennan, Lorraine (*n* = 1484)
Celis-Morales, Carlos (*n* = 33)	Navas-Carretero, Santiago (*n* = 1323)
Livingstone, Katherine M. (*n* = 31)	Gibney, Eileen R. (*n* = 1295)
Kang, Wenyi (*n* = 30)	San-Cristobal, Rodrigo (*n* = 1234)
Walsh, Marianne C. (*n* = 29)	Alfredo Martinez, J. (*n* = 1222)
Macready, Anna L. (*n* = 29)	Mozaffarian, Dariush (*n* = 1217)

## Data Availability

The original data presented in the study were obtained from Web of Science, and the methodological approach is described in the manuscript.

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
