# Peer review of "Precision or Personalized Nutrition: A Bibliometric Analysis"

_nutrients, 2024, doi:10.3390/nu16172922_

Round 1

Reviewer 1 Report

Comments and Suggestions for Authors

Thank you for invitation for review article tite: Precision or Personalized Nutrition: A Bibliometric Analysis.

My remarks:

Abstract should be improove (aim of the study, conclusion)

What was the aim of the study, please precise.

Pleace write more information about metodhology, who made analisis, what were the hipotesais

what were conclusions

Please develop limitation of the study.

Figure 2, 3a 3b  are not readible. please present this data another way.

Author Response

First of all, the authors would like to thank the anonymous reviewer for the helpful and constructive comments that greatly contributed to improving the final version of the manuscript. The authors have tried to follow and answer all the recommendations and commentaries of the reviewer. The answers of the authors have been indicated by “AU”. The changes within the manuscript have been highlighted in red font in the revised manuscript. Authors hope the revision will meet the demands of the reviewer and Editor.

Thank you for invitation for review article tite: Precision or Personalized Nutrition: A Bibliometric Analysis.

My remarks:

Abstract should be improve (aim of the study, conclusion)

AU: We would like to thank the reviewer for their comments, which have enabled us to make improvements to the manuscript. As suggested by the reviewer we have adapted the abstract with a more detailed exposition of the study's objective and conclusion.

What was the aim of the study, please precise.

AU: Thanks for suggestions about the objective and hypotheses of the study. We have adapted and clarified these issues within the manuscript (lines 77–83)

Please, write more information about methodology, who made analysis, what were the hypothesis

AU: We appreciate the comment of reviewer. The methodology information has been reorganized and improved in order to clarify the process and purpose. Additionally, with regard to the persons in charge of the analysis, this information is completely described in the section entitled "Author Contributions."

what were conclusions

AU: As requested by the reviewer, the conclusions of the study have been improved within its section.

Please develop limitation of the study.

AU: We would like to express our gratitude to the reviewer for his comments. Furthermore, although limitations of the study were already described in the original manuscript, we have improved this section following the recommendation of the reviewer (lines 388-396).

Figure 2, 3a 3b are not readible. please present this data another way.

AU: The figures have been amended to improve clarity. Figure 3 has been separated into figures 3 and 4 for a better view.

We would like to thank the reviewer for his comments and hope that the new version of the manuscript is more suitable for publication.

Reviewer 2 Report

Comments and Suggestions for Authors

Dear Authors,

The title of the manuscript (nutrients-3131172) submitted for review is quite interesting. The manuscript is innovative and interesting, as well as well-written.The introduction, material and methods, results, and discussion section are well-written. Despite this, I do not recommend it for publication. It seems to me that the article's purpose, i.e., bibliometric analysis without analysis of the content of the articles, has little value for other readers. We learn from the manuscript how many authors publish on this topic, from which institutions, and whether they are well cited.

The manuscript lacked information, some summary of what conclusions these articles concern, what the authors write about, and whether the methodologies they used are correct. What the authors found in their research. I believe the conclusions should include information for future researchers, what can still be done on this topic, and what has not been discovered. The authors spent a lot of time reviewing the article databases, but it seems that they should work on what they found, and such an article would undoubtedly be interesting and necessary.

Technical notes

The figures in the manuscript are outstanding, but the black background makes them illegible.

References:

References are cited according to journal rules.

 Despite my comments, I do not recommend this manuscript for publication.I believe the manuscript should be supplemented with a content analysis of the articles and re-submitted for evaluation.

 Reviewer

Author Response

First of all, the authors would like to thank the anonymous reviewer for the helpful and constructive comments that greatly contributed to improving the final version of the manuscript. The authors have tried to follow and answer all the recommendations and commentaries of the reviewer. The answers of the authors have been indicated by “AU”. The changes within the manuscript have been highlighted in red font in the revised manuscript. Authors hope the revision will meet the demands of the reviewer and Editor.

Dear Authors,

The title of the manuscript (nutrients-3131172) submitted for review is quite interesting. The manuscript is innovative and interesting, as well as well-written. The introduction, material and methods, results, and discussion section are well-written. Despite this, I do not recommend it for publication. It seems to me that the article's purpose, i.e., bibliometric analysis without analysis of the content of the articles, has little value for other readers. We learn from the manuscript how many authors publish on this topic, from which institutions, and whether they are well cited.

The manuscript lacked information, some summary of what conclusions these articles concern, what the authors write about, and whether the methodologies they used are correct. What the authors found in their research. I believe the conclusions should include information for future researchers, what can still be done on this topic, and what has not been discovered. The authors spent a lot of time reviewing the article databases, but it seems that they should work on what they found, and such an article would undoubtedly be interesting and necessary.

AU: We want to thank the reviewer for their comments and suggestions, and as we would like to change the reviewer's decision, we have sought to improve the new version of the manuscript with the help of the recommendations provided.

First of all, one of the strengths of the manuscript approach is precisely the different and innovative perspective of a bibliometric analysis. what they pointed out, approaching PN from a different and innovative perspective. In other fields, bibliometric analyses are widely used as an alternative to systematic reviews, providing a different type of information. Bibliometric analyses aim to provide insight into patterns of knowledge production and accumulation. For that purpose, individual studies are not examined in detail, as the focus of the analysis is not the synthesis of research findings per se, adding valuable information to the existing traditional reviews, something that we believe it is of interest to the readers.

However, we appreciate the comment of reviewer and as the reviewer indicated, we have added more information about the results obtained in PN interventions. Additionally, as the reviewer suggested, we have added a new paragraph describing future perspectives on this topic. All these modifications have been added within the discussion section.

Technical notes

The figures in the manuscript are outstanding, but the black background makes them illegible.

AU: Finally, we want to thank the reviewer for their technical notes. We would like to point out that we have improved the quality of all the figures, and despite the black background, we believe they are now clearer.

References:

References are cited according to journal rules.

AU: The authors thanks the reviewer’s comment.

 Despite my comments, I do not recommend this manuscript for publication. I believe the manuscript should be supplemented with a content analysis of the articles and re-submitted for evaluation.

AU: We want to thank the reviewer for their constructive comments, as they have allowed us to improve the scientific quality of the manuscript. We hope that this new version of the manuscript is much more interesting to the reviewer and that it will be considered for publication.

Reviewer 3 Report

Comments and Suggestions for Authors

Dear authors,

This systematic literature review clearly presents increasing trends in PN research and related fields, emphasizing potential trace for further nutrition research. I only have few comments on terminology/ english style and quality of figures.

Comments on the Quality of English Language

Just few style remarks, left in the text as comments.

Author Response

First of all, the authors would like to thank the anonymous reviewer for the helpful and constructive comments that greatly contributed to improving the final version of the manuscript. The authors have tried to follow and answer all the recommendations and commentaries of the reviewer. The answers of the authors have been indicated by “AU”. The changes within the manuscript have been highlighted in red font in the revised manuscript. Authors hope the revision will meet the demands of the reviewer and Editor.

Dear authors,

This systematic literature review clearly presents increasing trends in PN research and related fields, emphasizing potential trace for further nutrition research. I only have few comments on terminology/ english style and quality of figures.

AU: We would like to acknowledge the reviewer's comments as a constructive contribution to the process of improving the quality of the manuscript. Furthermore, we consider the enhancements made to the English terminology and style to be highly beneficial in enhancing the clarity of the manuscript. All recommendations made by the reviewer have been implemented throughout the document.

With regard to the quality of the figures, the reviewer's assessment is accurate. We have elected to enhance the contrast of Figure 2 and have divided Figure 3a and b into Figures 3 and 4, respectively, while also increasing the contrast.

We consider this updated version of the manuscript to be more coherent than the previous one. We would like to express our sincerest gratitude.

Reviewer 4 Report

Comments and Suggestions for Authors

Dear Authors

The paper is really interesting and has an original approach to the topic . It is a deep analysis of published papers on precision or personalised nutrition during the time.

I congratulate the authors for this well-written article.

Please find some minor comments below:

I think Figure 1 can  be split into two figures one with published papers by years and the other countries by documents and citations.

It would be more understandable if you could write the explanations (lines 177-188) for Figures 3A and 3B above the Figure.

Kind regards

Author Response

First of all, the authors would like to thank the anonymous reviewer for the helpful and constructive comments that greatly contributed to improving the final version of the manuscript. The authors have tried to follow and answer all the recommendations and commentaries of the reviewer. The answers of the authors have been indicated by “AU”. The changes within the manuscript have been highlighted in red font in the revised manuscript. Authors hope the revision will meet the demands of the reviewer and Editor.

Dear Authors

The paper is really interesting and has an original approach to the topic . It is a deep analysis of published papers on precision or personalised nutrition during the time.

I congratulate the authors for this well-written article.

AU: We would like to express our gratitude to the reviewer with regard to their positive comments.

Please find some minor comments below:

I think Figure 1 can  be split into two figures one with published papers by years and the other countries by documents and citations.

AU: In response to the recommendation to divide Figure 1 into two, after carefully trying this and other approaches, we have decided to retainer it as a single entity. The rationale for this decision is that the information is readily discernible and that the current format is more optimal in terms of the efficiency of the data presentation.

It would be more understandable if you could write the explanations (lines 177-188) for Figures 3A and 3B above the Figure.

AU: On the other hand, in accordance with the reviewer's recommendation, the figures have been relocated and divided, thus facilitating greater clarity.

We consider this updated version of the manuscript to be more coherent than the previous one. We would like to express our sincerest gratitude.

Round 2

Reviewer 2 Report

Comments and Suggestions for Authors

Dear Authors,

The authors have added quite extensive fragments of their manuscript, but these are still mainly statistical data and not information important for future readers/scientists. I would rather see this as the main idea presented in these articles, as it is presented in lines 327-329.

 Lines 327-329 are the kind of analysis I would like to see in this work. Maybe the authors will put some effort into this article because it is well-written but lacks that WOW effect.

 I suggest preparing a table from this text:

1 column: population, e.g., general;

2 column: dietary intervention, e.g., mobile applications,

3 column: main conclusion;

4 column: e.g., Authors [65].

It would clearly summarize the results found in the analysis (I hope that analyzed and not only prepared statistics data) articles. I have the impression that authors only used the database of articles without examining their content,

 Technical notes: Figure 1 still has a tiny, illegible font on the circles.

I leave the decision to publish this article to the Editors.

Reviewer

Author Response

The authors would like to express their gratitude to the reviewer for their valuable comments. The authors have made every effort to address and incorporate the reviewer's recommendations and comments into the revised manuscript. The authors' responses have been indicated with "AU." The revised manuscript has been updated to reflect the new suggested changes, which have been highlighted in blue font. We hope that the revised manuscript meets the expectations of the reviewer and the editor.

Dear Authors,

The authors have added quite extensive fragments of their manuscript, but these are still mainly statistical data and not information important for future readers/scientists. I would rather see this as the main idea presented in these articles, as it is presented in lines 327-329.

 Lines 327-329 are the kind of analysis I would like to see in this work. Maybe the authors will put some effort into this article because it is well-written but lacks that WOW effect.

AU: We want to thank the reviewer for their comments and suggestions. We appreciate the fact that the reviewer has considered that the manuscript has been improved in this version. However, we would like to clarify that this article is a bibliometric analysis, meaning that it is a statistical analysis of literature, and although very interesting results in the manner of mainly trends are found, as the reviewer correctly notes, this is not a review article and being this statistic the fundamental objective of the manuscript. Indeed, to provide further context and demonstrate the rigor of our approach, we offer the reviewer a few examples of similar analyses.

Lin, X.; Wang, S.; Huang, J. A Bibliometric Analysis of Alternate-Day Fasting from 2000 to 2023. Nutrients 202315, 3724. https://doi.org/10.3390/nu15173724

Gutiérrez-Hellín, J.; Del Coso, J.; Espada, M.C.; Hernández-Beltrán, V.; Ferreira, C.C.; Varillas-Delgado, D.; Mendoza Laiz, N.; Roberts, J.D.; Gamonales, J.M. Research Trends in the Effect of Caffeine Intake on Fat Oxidation: A Bibliometric and Visual Analysis. Nutrients 202315, 4320. https://doi.org/10.3390/nu15204320

We hope that this will help to illustrate that our study aligns closely with the established methodology.

 I suggest preparing a table from this text:

1 column: population, e.g., general;

2 column: dietary intervention, e.g., mobile applications,

3 column: main conclusion;

4 column: e.g., Authors [65].

It would clearly summarize the results found in the analysis (I hope that analyzed and not only prepared statistics data) articles. I have the impression that authors only used the database of articles without examining their content,

AU: The authors thank the reviewer this suggestion in order to improve the current manuscript. However, after a full consideration of the reviewer’s recommendation by the authors, and in light of the aforementioned considerations, the table we have found that the mentioned table should not be considered as a fundamental component of the article. Given that this manuscript is not a literature review, the inclusion of the table of some articles and not others could result arbitrary, as the articles cited in the mentioned paragraph should be considered as examples. However, taking into account the interest of the reviewer in this part of the manuscript, we have decided to improve it by adding more information about relevant facts for the readers (lines 333-337; lines 371-375 and lines 382-390).

 Technical notes: Figure 1 still has a tiny, illegible font on the circles.

AU: We would like to express our gratitude to the reviewer for offering constructive feedback on the figure. In an effort to enhance the visual representation, the font size of the text within the circles has been increased. We believe that the new figure provides a more optimal visual representation.

I leave the decision to publish this article to the Editors.

AU:  The authors are hopeful that this revised version of the manuscript will be considered for publication and will meet the standards required for its inclusion in the journal.